# In situ copper photocatalysts triggering halide atom transfer of unactivated alkyl halides for general C(sp3)-N couplings

Hang Luo[1,4], Yupeng Yang[2,4], Yukang Fu[3,4], Fangnian Yu[1], Lei Gao[1], Yunpeng Ma[1], Yang Li ![ORCID] [3] ✉, Kaifeng Wu ![ORCID] [2] ✉ & Luqing Lin ![ORCID] [1] ✉

Direct reduction of unactivated alkyl halides for C(sp3)-N couplings under mild conditions presents a significant challenge in organic synthesis due to their low reduction potential. Herein, we introduce an in situ formed pyridyl-carbene-ligated copper (I) catalyst that is capable of abstracting halide atom and generating alkyl radicals for general C(sp3)-N couplings under visible light. Control experiments confirmed that the mono-pyridyl-carbene-ligated copper complex is the active species responsible for catalysis. Mechanistic investigations using transient absorption spectroscopy across multiple decades of timescales revealed ultrafast intersystem crossing (260 ps) of the photo-excited copper (I) complexes into their long-lived triplet excited states (>2 µs). The non-Stern-Volmer quenching dynamics of the triplets by unactivated alkyl halides suggests an association between copper (I) complexes and alkyl halides, thereby facilitating the abstraction of halide atoms via inner-sphere single electron transfer (SET), rather than outer-sphere SET, for the formation of alkyl radicals for subsequent cross couplings.

Nitrogen-containing compounds are commonly found in natural products, pharmaceuticals, and agrochemicals[1–3]. The C-N coupling reaction is a significant method for the rapid assembly of nitrogenated compounds. Transition-metal-catalyzed C(sp2)-N couplings have been well developed and widely applied in industry[4,5]. For the construction of C(sp3)-N bonds, nucleophilic substitution (SN1 or SN2) reactions are useful and simple methods, but electrophiles are typically limited to activated alkyl halides and their derivatives. For thermochemical reactions, several excellent works including asymmetric C(sp3)-N coupling have been well developed between activated alkyl halides and amines, but they are still not suitable for alkylation of amines using unactivated alkyl halides[6–9].

Recently, photocopper catalysis has broadened the avenue for formal substitution reactions to couple unactivated alkyl halides with diverse N-nucleophiles via a single electron transfer (SET) process

(Fig. 1a)[10,11]. However, the reaction conditions for these couplings are not generic, requiring changes to parameters such as ligands, light, and solvent, and even requiring co-ligands for coupling with diverse N-nucleophiles[12–16]. To achieve the desired redox potentials for unactivated alkyl halides ($E_{red} < -2.0$ V vs SCE), accessing highly reducing copper species is crucial. Fu, Peters, and their colleagues have previously reported the use of highly reducing excited Cu(I)-amido complexes (amido = carbazole, indole, or amide) for generating alkyl radicals. However, these methods were only applicable to a specific type of N-nucleophiles[12–14]. As a follow-up, they discovered new P-N-P/P-P ligand ligated photo-copper catalysts that are not Cu(I)-amido complexes for couplings[15,16]. Nonetheless, the choice of N-nucleophile remains limited[15,16]. Despite significant progress being made in copper catalysis, the low reduction potential of unactivated secondary alkyl halides[17] remains a major obstacle that hampers the further

[1]School of Chemistry, Dalian University of Technology, Dalian, Liaoning 116024, China. [2]State Key Laboratory of Molecular Reaction Dynamics and Collaborative Innovation Center of Chemistry for Energy Materials (iChEM), Dalian Institute of Chemical Physics, Chinese Academy of Sciences Dalian, Dalian, Liaoning 116023, China. [3]State Key Laboratory of Fine Chemicals, School of Chemical Engineering, Dalian University of Technology, Dalian, Liaoning 116024, China. [4]These authors contributed equally: Hang Luo, Yupeng Yang, Yukang Fu. ✉e-mail: chyangli@dlut.edu.cn; kwu@dicp.ac.cn; linluqing@dlut.edu.cn

**a** General C(sp3)-N couplings in copper catalysis

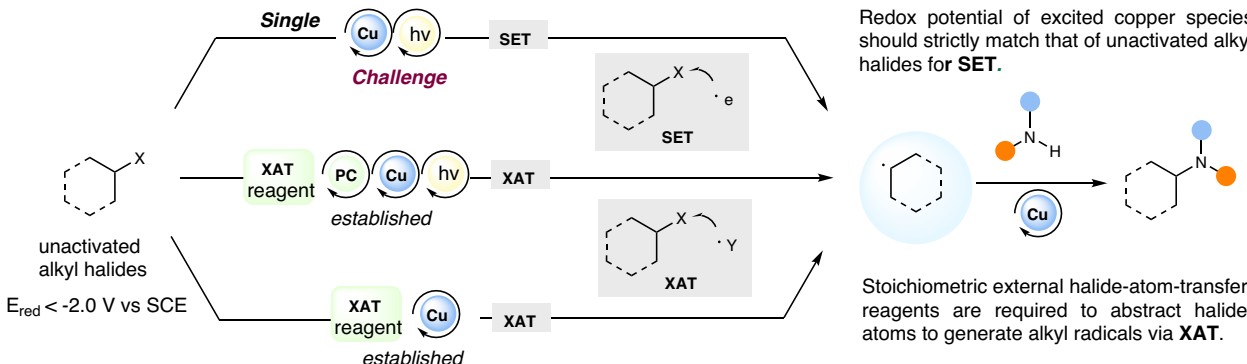

Redox potential of excited copper species should strictly match that of unactivated alkyl halides for **SET**.

Stoichiometric external halide-atom-transfer reagents are required to abstract halide atoms to generate alkyl radicals via **XAT**.

**b This work:** "One-stone-for-two-birds" copper catalysis for general C-N couplings via excited-copper-catalytic XAT process

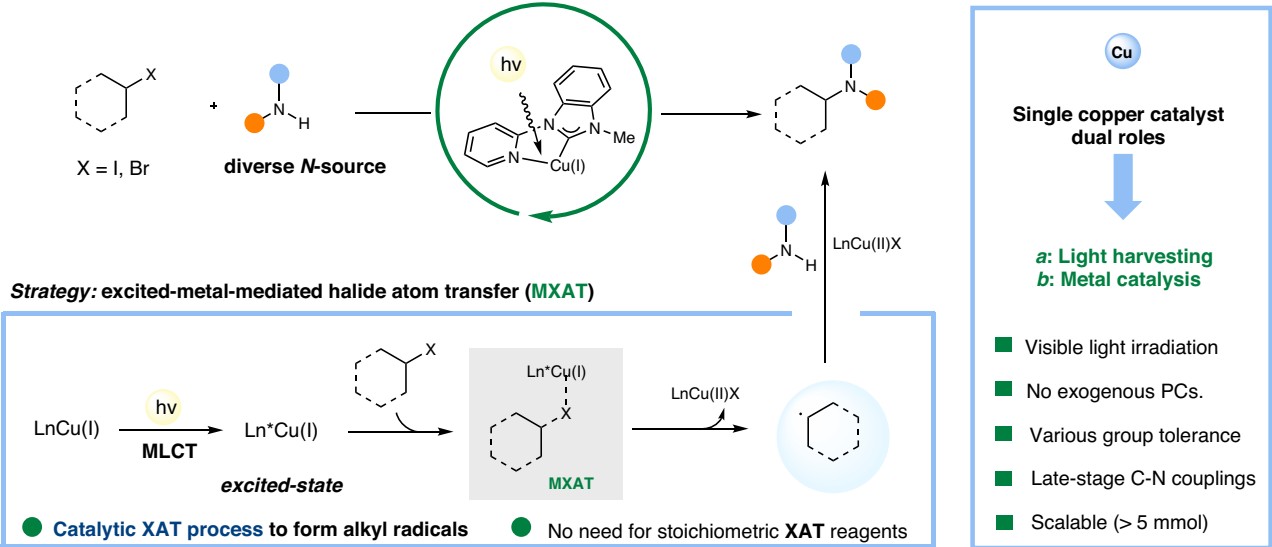

**Fig. 1 | Different strategies to achieve copper-catalyzed C-N couplings with unactivated alkyl halides. a** Diverse strategies for copper-catalysed C(sp3)-N couplings. **b** Photo-induced pyridyl-carbene-ligated copper (I) catalyst for general C-N couplings. SET single electron transfer. XAT halide atom transfer.

development of general methods for coupling diverse N-substrates and unactivated alkyl halides via SET process.

On the other hand, while various general C-N couplings utilizing halide atom transfer (XAT) strategies to produce alkyl radicals from alkyl halides have been developed successfully, the excessive utilization of XAT reagents creates a significant amount of byproduct in the XAT process, which poses a challenge in terms of sustainability[18,19]. (Fig. 1a). Other methods utilizing carboxylic acid and its derivatives as alkyl radical sources have also been accomplished[20,21]; however, none of them can be extended to alkyl halides. Thus, developing a general copper-catalyzed C(sp3)-N coupling of unactivated alkyl halides with diverse N-substrates that is not specific to a particular type of N-nucleophile remains a challenge.

Overall, while copper complexes can serve as both photocatalysts and metal catalysts, which helps streamline the reaction process[22], many copper(I) photocatalysts[11,23,24] do not possess the right redox potentials required to efficiently reduce unactivated alkyl halides through an outer-sphere SET process for general C-N couplings. Different from the outer-sphere SET process, which requires a strict match of redox potentials for alkyl halides, metal mediated atom transfer process via the inner-sphere SET pathway seems to not require reducing metal species with fully matched redox potentials. Instead, it somehow relies on halogenophilicity of the excited-state copper, potentially offering an alternative method to access alkyl radicals for

general C-N couplings[25–27]. This strategy would realize catalytic XAT process to from alkyl radical avoiding the use of stoichiometric XAT reagents and exogenous photocatalysts.

In this report, we unveil a copper catalyst ligated with a pyridyl-carbene for abstracting halide atoms of unactivated alkyl iodides/bromides via inner-sphere pathway under visible-light irradiation (Fig. 1b)[28,29]. This in situ catalyst enables access to C(sp3)-N couplings, providing a versatile N-alkylation platform that allows for the coupling of unactivated alkyl halides with various N-nucleophiles under mild conditions. The practicality of this methodology is demonstrated through both scale-up reaction and late-stage modification of complex molecules.

## Results
### Reaction development and substrate scope
Based on the reported structure of copper photocatalysts[11], we assumed that a copper complex ligated with an electron-rich pyridyl-carbene ligand could potentially enable the unexplored alkylation of anilines with unactivated iodides under photo-irradiation. After carefully screening various parameters, we eventually identified the optimal conditions, which entailed the use of copper acetate (5 mol%) as the metal source, Py-NHC compound (**L1**, 5 mol%) as the ligand precursor, and 7-methyl-1,5,7-triazabicyclo[4.4.0]dec-5-ene (MTBD) as the base in acetonitrile under 410 nm LEDs irradiation at room

**Table 1 | Optimization of conditions for C-N couplings[a]**

| Entry | Variants from standard condition | Yield (%)[b] |
|---|---|---|
| 1 | none | 94(93)[c] |
| 2 | **L2** instead of **L1** | 94 |
| 3 | **L3** instead of **L1** | 75 |
| 4 | **L4** instead of **L1** | 91 |
| 5 | **L5** instead of **L1** | trace |
| 6 | *t*BuOLi instead of MTBD | 47 |
| 7 | BTMG instead of MTBD | 30 |
| 8 | DBU instead of MTBD | N.D. |
| 9 | DIPEA instead of MTBD | N.D. |
| 10 | DMF instead of $CH_3CN$ | 70 |
| 11 | THF instead of $CH_3CN$ | 80 |
| 12 | MeOH instead of $CH_3CN$ | N.D. |
| 13 | CuI instead of CuOAc | 93 |
| 14 | CuBr instead of CuOAc | 93 |
| 15 | CuCl instead of CuOAc | 30 |
| 16 | $Cu(CH_3CN)_4PF_6$ instead of CuOAc | 20 |
| 17 | Without **L1** | N.D. |
| 18 | In dark | N.D. |
| 19 | Without MTBD or CuOAc | N.D. |

*N.D.* not detected, *DIPEA* N,N-Diisopropylethylamine, *DMF* N,N-Dimethylformamide.
[a]Reactions were run at 0.1 mmol scale.
[b]The yield was determined by [1]H NMR using 1,3,5-trimethoxybenzene as the internal standard.
[c]Isolated yield in parentheses.

temperature. Under these conditions, we successfully coupled the unactivated *tert*-butyl 4-iodopiperidine-1-carboxylate **1** with the *p*-toluidine **2**, thereby delivering the desired product **3** in 93% isolated yield (Table 1, Entry 1). To elucidate the significance of each element, we conducted several control experiments. Notably, the electron-rich methyl group on the pyridine ring displayed comparable reactivity to **L1**, while the electron-withdrawing group trifluoromethyl group significantly reduced the reactivity (Entries 2 & 3). The use of N-*para*-trifluoromethyl phenyl instead of the N-methyl group slightly decreased the yield (Entry 4). Substituting the pyridine ring with quinoline completely suppressed the reaction (Entry 5). These results indicated that the electron-rich pyridine moiety enhances the reactivity of the in-situ generated copper complex. Furthermore, several common bases were evaluated, including the strong inorganic base lithium *tert*-butoxide, which was found to have poor reactivity and caused serious unwanted side reactions, such as β-hydrogen elimination and deiodination (Entry 6 and Supplementary Table 1). Among the organic bases tested, only 2-*tert*-butyl-1,1,3,3-tetramethylguanidine

(BTMG) was able to promote the reaction, but its reactivity was still not as good as that of MTBD (Entries 7-9). Solvents including DMF, THF, and MeOH did not provide better results (Entries 10–12). The copper source showed diverse reactivity, indicating that the associated anion is also a significant factor (Entries 13-16). Control experiments (Entries 17–19) confirmed that the presence of the ligand, light, MTBD, and copper was all necessary for the reaction to proceed. The 1,10-phenanthroline copper complex, which is commonly employed in atom transfer radical addition reactions[11], failed to yield any product (Supplementary Table 4, Entry 20).

Various N-nucleophiles were subjected to the optimal conditions for C(sp3)-N couplings in this study (Fig. 2). The catalytic process effectively accommodated both electron-rich and electron-poor anilines (**3-8**). Notably, the steric hindrance of N-nucleophiles did not significantly impact the reaction system, as demonstrated by the successful coupling of anilines with mono-substitution or disubstitution on *ortho*-positions (**5-7**). In addition, the bromide and cyano groups were found to be highly suitable for further functionalization and were

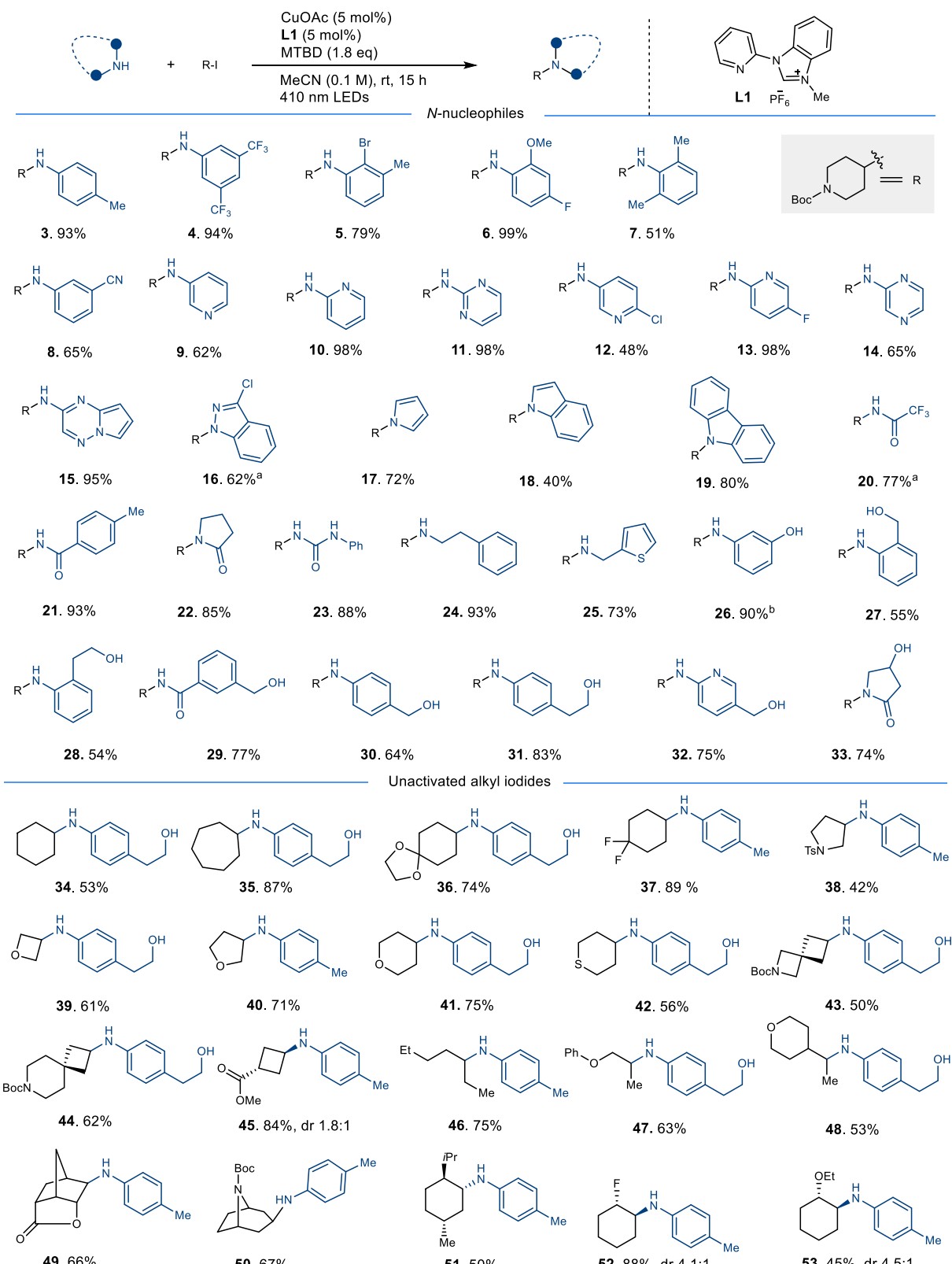

**Fig. 2 | The substrate scope of N-nucleophiles and alkyl iodides.** The reactions were run using N-nucleophiles (0.1 mmol), alkyl iodides (0.15 mmol). [a]*t*BuOLi instead of MTBD, [b]Using MTBD (0.28 mmol).

compatible under the experimental conditions (**5, 8**). The successful N-alkylation of various important heterocycles such as pyridines, pyrazine, pyrimidine, indole, pyrrole, carbazole, and indazole (**9-19**) provided a reliable pathway for accessing crucial pharmaceutical moieties. Furthermore, both cyclic and acyclic amides and carbamates

were effectively transformed into C(sp3)-N coupled products (**20-23**). Notably, a urea was also successfully coupled with an alkyl iodide (**23**). It is worth noting that under the present conditions, primary amines were selectively mono-alkylated to form the desired products (**24-25**), and over-alkylation products were not observed. Secondary amines

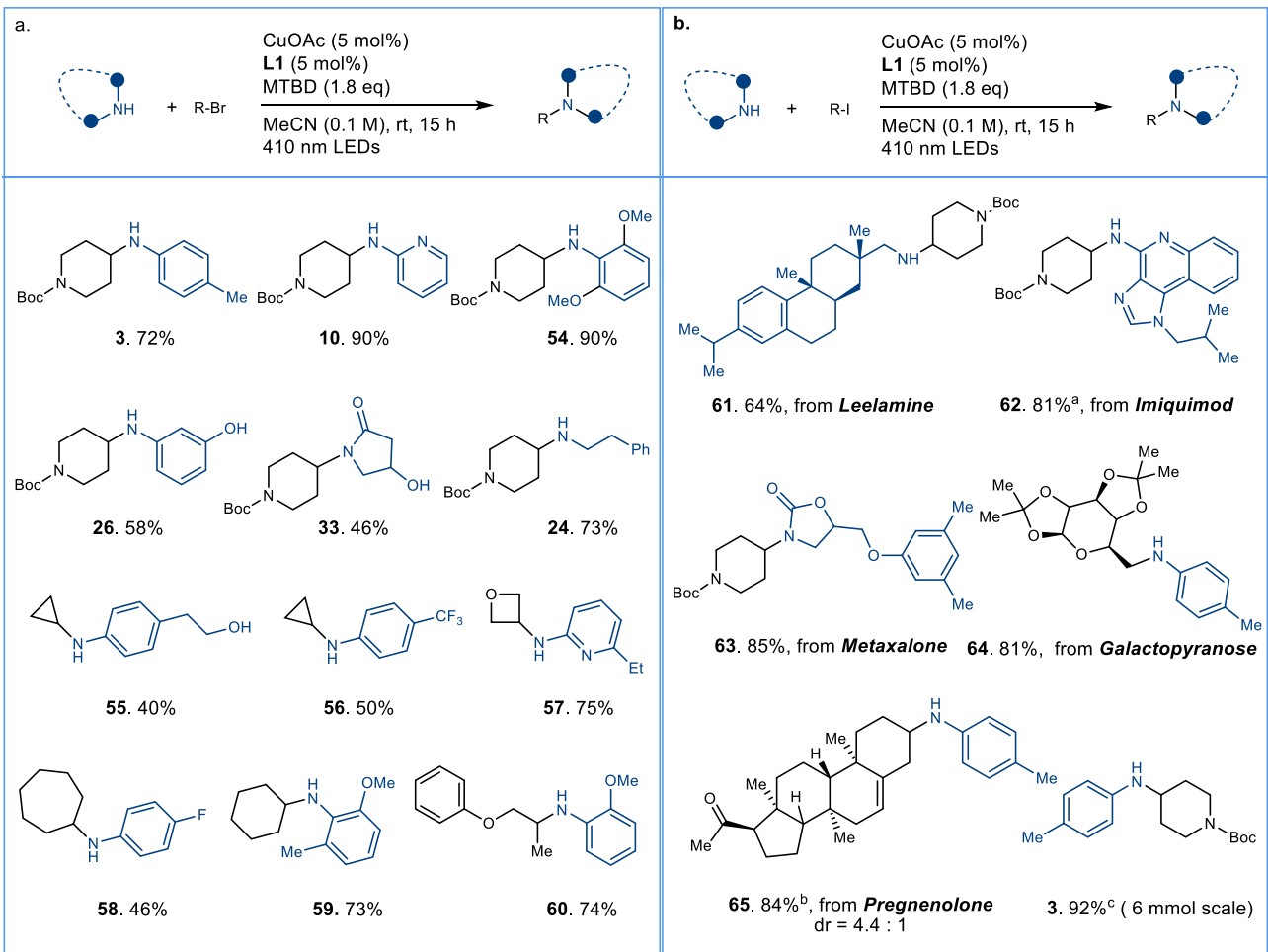

**Fig. 3 | Extended application of pyridyl-carbene copper catalysis. a** The substrate scope of amination of alkyl bromides. **b** Late-stage modification of complex molecules and the scale-up reaction. The reactions were run using N-nucleophiles (0.1 mmol) and alkyl bromides/iodides (0.15 mmol) under standard conditions unless otherwise noted. [a]using DMF/CH₃CN = 9/1 as the solvent, [b]using PhCF₃/CH₃CN = 9/1 as the solvent, [c]18 h.

were ineffective, possibly because of the slow deprotonation process hindering their coordination with copper via ligand substitution under the current conditions. (Supplementary Fig. 24).

The free hydroxyl group is a highly valuable functional moiety present in numerous natural products, pharmaceuticals and materials[30]. However, due to its susceptibility to side reactions during chemical transformations, it often necessitates meticulous protection and deprotection procedures. Therefore, achieving selective alkylation on nitrogen atoms in substrates containing a free hydroxyl group is of paramount importance. In this context, we evaluated the efficacy of such substrates, including those bearing a phenol ring (**26**). Encouragingly, these substrates were well-tolerated and efficiently converted into the desired products (**26-33**; yields ranging from 54-90%), without any occurrence of undesirable C-O coupling side products. Notably, the bi-dentate N,O-ligand, which has the potential to deactivate the copper catalyst in the presence of a strong base, also yielded the desired product (**27**) in moderate yield.

Next, we sought to expand approach on the substrate scope of alkyl iodides. We opted for electron-rich nucleophilic anilines that were previously incompatible in copper-catalyzed alkylation with unactivated alkyl halides (Fig. 2)[19,20]. Our investigation showed that both cyclic and acyclic iodides exhibited good reactivity that efficiently led to the formation of C-N coupling products (**34-53**). Furthermore, our findings demonstrated that even strained heterocyclic substrates could be readily converted into their corresponding coupled products in good yields (**39, 43-45**). Of particular interest, some

highly useful spiro and bridged bicyclic fragments were also successfully incorporated into the amination process (**43, 44**). Additionally, functionalized iodides, such as protected ketones, esters, and protected amines, have been demonstrated to exhibit exceptional performance in the formation of coupled products (**36, 43-45, 50**). Interestingly, iodolactone is able to proceed efficiently while maintaining the integrity of its lactone ring (**49**). Furthermore, our results demonstrated that the presence of vicinal fluoro, ethoxy, or bulky isopropyl groups on the substrates favored amination from the less sterically hindered side, leading to the formation of the desired products (**51-53**).

We subsequently made efforts to broaden the protocol to facilitate the coupling reaction between alkyl bromides and various N-nucleophiles (Fig. 3a). Gratifyingly, a variety of N-nucleophiles, including anilines, amides, and primary amines, were efficiently alkylated using cyclic or acyclic alkyl bromides as depicted in Fig. 3. Additionally, nucleophiles featuring free hydroxyl groups, such as those found in compounds **26, 33**, and **55**, were suitable substrates for alkylation. Even sterically hindered anilines were amenable to alkylation, furnishing the desired products **54** and **59** in high yield. Notably, cyclopropyl bromide was also demonstrated to serve as a competent alkylation reagent, yielding the coupled products **55** and **56** in moderate yield. In contrast, attempts to achieve the coupling of alkyl chlorides with N-nucleophiles proved unsuccessful under the present conditions. However, other electrophiles with higher redox potential, such as redox esters, are incompatible (Supplementary Fig. 27).

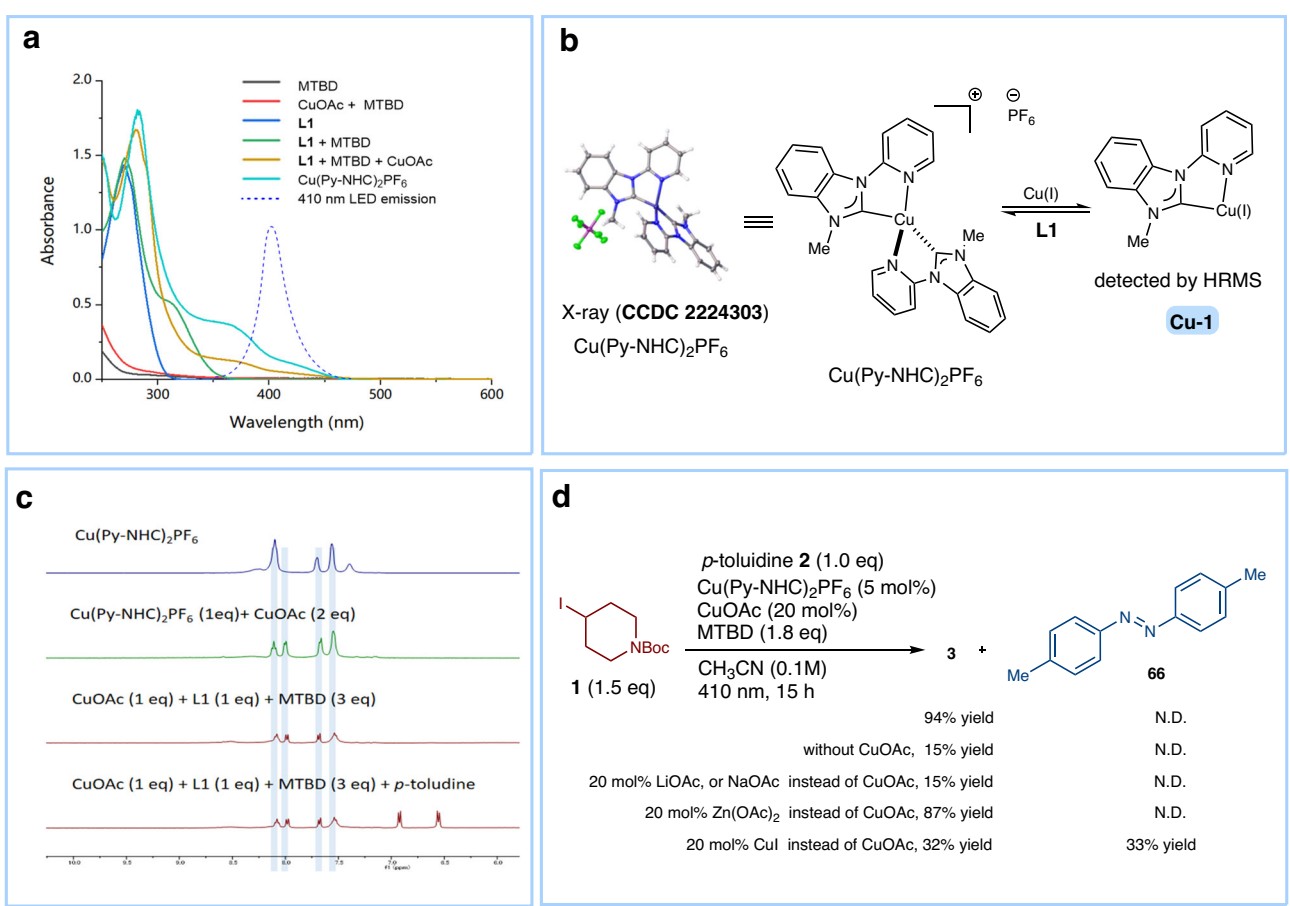

**Fig. 4 | Exploration of the active copper species. a** UV-vis spectra and emission spectra of LED light. **b** Cu(Py-NHC)$_2$PF$_6$ complex in solution. **c** $^1$H NMR spectra of copper complexes. **d** Investigation into the active Cu$^I$ species involved in the photochemical process.

To further assess the efficacy of the C-N coupling protocol, we evaluated the reactivity of several complex molecular derivatives using the present conditions. As anticipated, coupling reactions proceeded smoothly for complex alkyl iodides or N-nucleophiles, resulting in good yield of coupling products (**61-65**). We then proceeded to assess the feasibility of scaling up the C-N cross-coupling reaction. The reaction performed well on a gram-scale (6 mmol) under the current conditions, yielding the final product **3** (1.6 g) in 92% yield over 18 h. These outcomes showcase the benefits of the optimized method and its potential application in organic synthesis (Fig. 3b). Attempts to achieve enantioselective C-N couplings[6–9,16,31–35] were unsuccessful with chiral Py-NHC ligands, possibly because of the lack of efficient chiral ligands to differentiate two similar methylene groups. (Supplementary Fig. 28).

## Identification of the catalyst species

To identify the catalytic species for the C-N coupling process, several experiments were conducted (Fig. 4). The UV-vis absorbance spectra showed that the in-situ generated pyridyl-carbene ligated copper complex (yellow line) has two predominant transitions: a π → π* band below 320 nm and a lower energy band at >350 nm assigned to metal-to-ligand charge transfer (MLCT) or ligand-to-ligand charge transfer[24,36]. Thus, the copper complex can be excited by 410 nm LEDs (blue dashed line) (Fig. 4a). We attempted to obtain the pyridyl-carbene copper complexes ex-situ, which would be beneficial for investigating and understanding the photocatalytic process. However, based on a 1:1 molar ratio of CuOAc to **L1**, we failed to obtain the three-coordinated Cu(Py-NHC)OAc complex. Instead, we were able to successfully synthesize and crystallize the Cu(Py-NHC)$_2$PF$_6$ complex

confirmed by X-ray analysis due to its good stability, similar to other tetracoordinated copper(I) complexes[24] (Fig. 4b). The solid Cu(Py-NHC)$_2$PF$_6$ complex has been found to be stable in air for at least three days. However, the Cu(Py-NHC)$_2$PF$_6$ complex dissolved in acetonitrile is extremely sensitive to air. We compared the cyclic voltammogram (CV) of Cu(Py-NHC)$_2$PF$_6$ to in situ generated copper complex and found them to have different redox potentials (0.42 vs SCE for the former & 0.29 V vs SCE for the latter). Thus, it is unlikely that the Cu(Py-NHC)$_2$PF$_6$ complex is the actual copper catalyst species in solution. Indeed, we discovered that the pure Cu(Py-NHC)$_2$PF$_6$ complex tends to lose one ligand when dissolved in solution, forming a mono-Py-NHC-ligated Cu(I) complex (**Cu-1**)[37,38]. This was confirmed by HRMS analysis (Fig. 4b and Supplementary Figs. 6 and 7). Furthermore, in the presence of excess copper acetate in acetonitrile, the Cu(Py-NHC)$_2$PF$_6$ complex can be readily converted into a new copper specie, as evidenced by its $^1$H NMR spectra which is identical to the mixture of CuOAc, **L1** and MTBD (Fig. 4c).

Several control experiments were further conducted to investigate the actual active Cu(I) species (Fig. 4d). The reactivity of Cu(Py-NHC)$_2$PF$_6$ towards C-N coupling was found to be poor, but the addition of excess CuOAc significantly enhanced the reactivity. One possible explanation for these results is that the [Cu$^I$(Py-NHC)$_2$] complex releases one Py-NHC ligand to form a potentially active **Cu-1** species. Interestingly, no positive results were obtained when CuOAc was replaced by LiOAc and NaOAc, suggesting that Cu$^+$ is an essential component for the formation of the new copper species. In contrast, replacing CuOAc with Zn(OAc)$_2$ notably increased reactivity, likely because Zn$^{2+}$ binds to the free Py-NHC ligand, promoting the formation of **Cu-1**. Replacing CuOAc with CuI led to the generation of a side

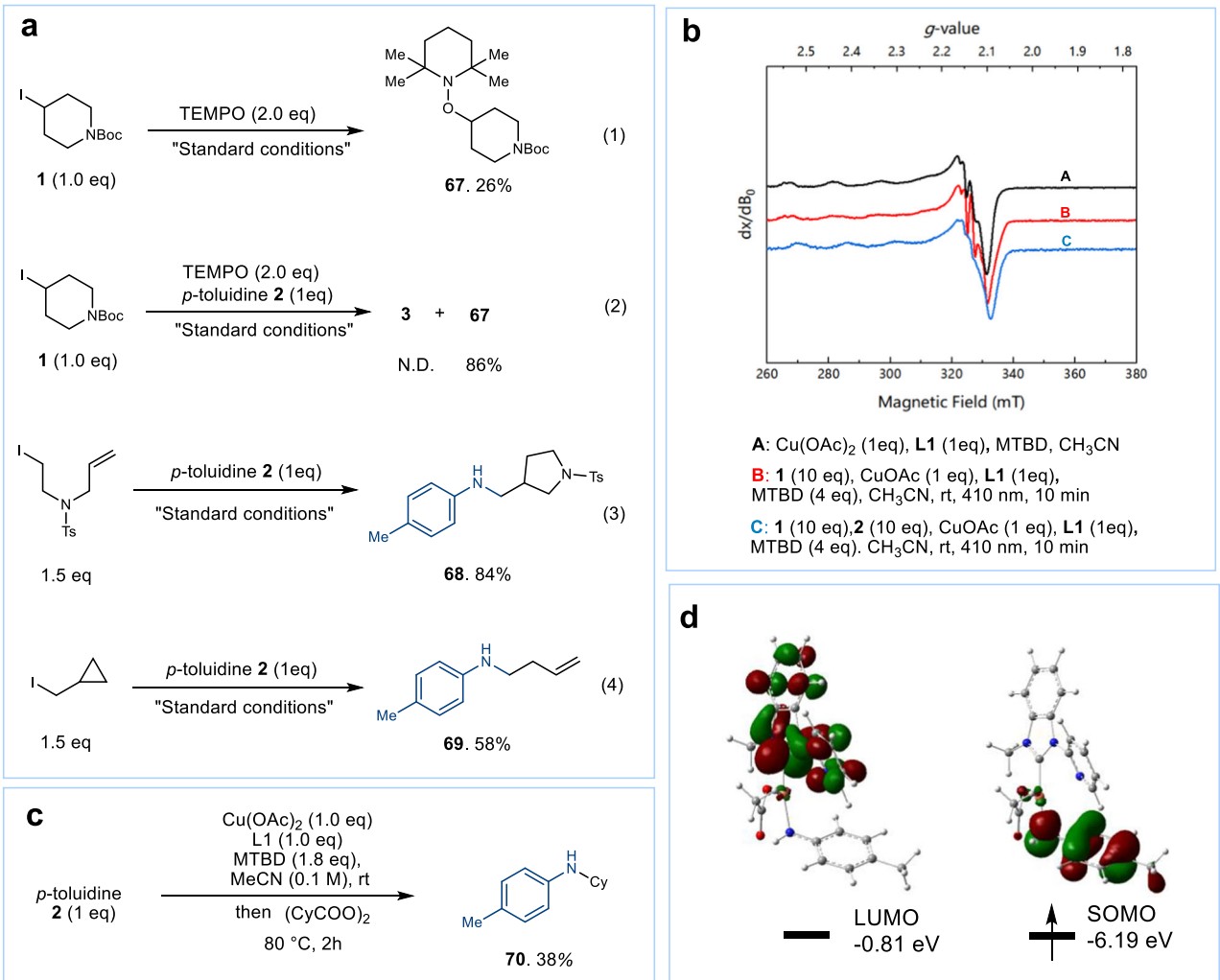

**Fig. 5 | Investigation of radical process of C-N couplings. a** Radical trapping experiments. **b** EPR spectra (X-band, 9.4 GHz, 100 K). Black trace: spectrum of mixture of Cu(OAc)$_2$, **L1**, and MTBD. Red trace: spectrum of mixture of modified catalytic condition without *p*-toluidine 2. Blue trace: spectrum of mixture of modified catalytic condition. **c** Investigation of Cu$^{II}$ intermediate for C-N coupling. **d** The lowest unoccupied molecular orbitals and singly occupied molecular orbitals of Cu$^{II}$(Py-NHC)(OAc)(NH-*p*-Tol), with isovalues of 0.04 a.u.

product **66**, likely because **Cu-1** becomes coordinated with *p*-toluidine in the absence of acetate anions, resulting in the oxidation of aniline to diazene that competes with the C-N coupling reaction[30,36]. In general, the presence of acetate anions and Cu$^+$ or Zn$^{2+}$cations that can bind to free Py-NHC ligands, inhibits the formation of Cu(Py-NHC)$_2$PF$_6$ and promotes the generation of **Cu-1**, accounting for all observed outcomes. Furthermore, aniline did not change the UV-Vis and $^1$H NMR spectra of the in-situ generated **Cu-1** (Fig. 4c and Supplementary Fig. 20). Therefore, it is expected that the **Cu-1** complex binds to the acetate anion rather than aniline to form Cu$^I$(Py-NHC)OAc as the photoactive complex in the reaction mixture.

**Investigation of radical process**

The active copper species identified above is capable of abstracting halide atoms from alkyl halides, leading to the formation of alkyl radicals through a SET mechanism. To confirm this radical process, we conducted radical trapping experiments under optimal conditions using various substrates (Fig. 5a). In Fig. 5a, we observed the formation of a TEMPO adduct **67** in the presence or absence of anilines under standard conditions, which was in line with our expectations (Fig. 5a-1, 2). Additionally, when using an iodoalkene substrate, a cyclic product **68** was formed (Fig. 5a-3). Furthermore, the radical clock reaction

resulted in formation of the C-C cleavage product **69** (Fig. 5a-4). These experimental results provide evidence supporting the existence of radical process.

EPR experiments were conducted to identify the Cu(II) complex if the catalytic process involves SET from Cu(I) to alkyl halides. Under standard conditions, the typical copper (II) signals (Fig. 5b, red and blue), as well as those observed without *p*-toluidine, were clearly detected after 10 min of irradiation, which are similar to the EPR spectrum of the Cu$^{II}$(Py-NHC)(OAc)$_2$ complex prepared in situ as a control example (Fig. 5b, black). To verify that the copper (II) complex mediates the C-N coupling, we conducted a stoichiometric experiment using peroxides as a radical source in the presence of the [Cu$^{II}$(Py-NHC)] complex generated in situ. This reaction led to the formation of a coupling product **70**, suggesting the participation of the copper(II) complex in the C-N coupling process (Fig. 5c). Several groups recently have demonstrated copper(II) mediated C-N formation through the coupling of anilidyl radical with the alkyl radical[39-42]. If the same process also occurred in our catalytic cycle, Cu$^{II}$(Py-NHC) should be reduced by the amine to form Cu(I) via an in-situ generated Cu$^{II}$-amido complex. Indeed, the EPR signal of the Cu$^{II}$(Py-NHC) complex (Fig. 5b, black) disappeared after the addition of *p*-toluidine for 60 min (Supplementary Fig. 9). Furthermore, Density Functional Theory

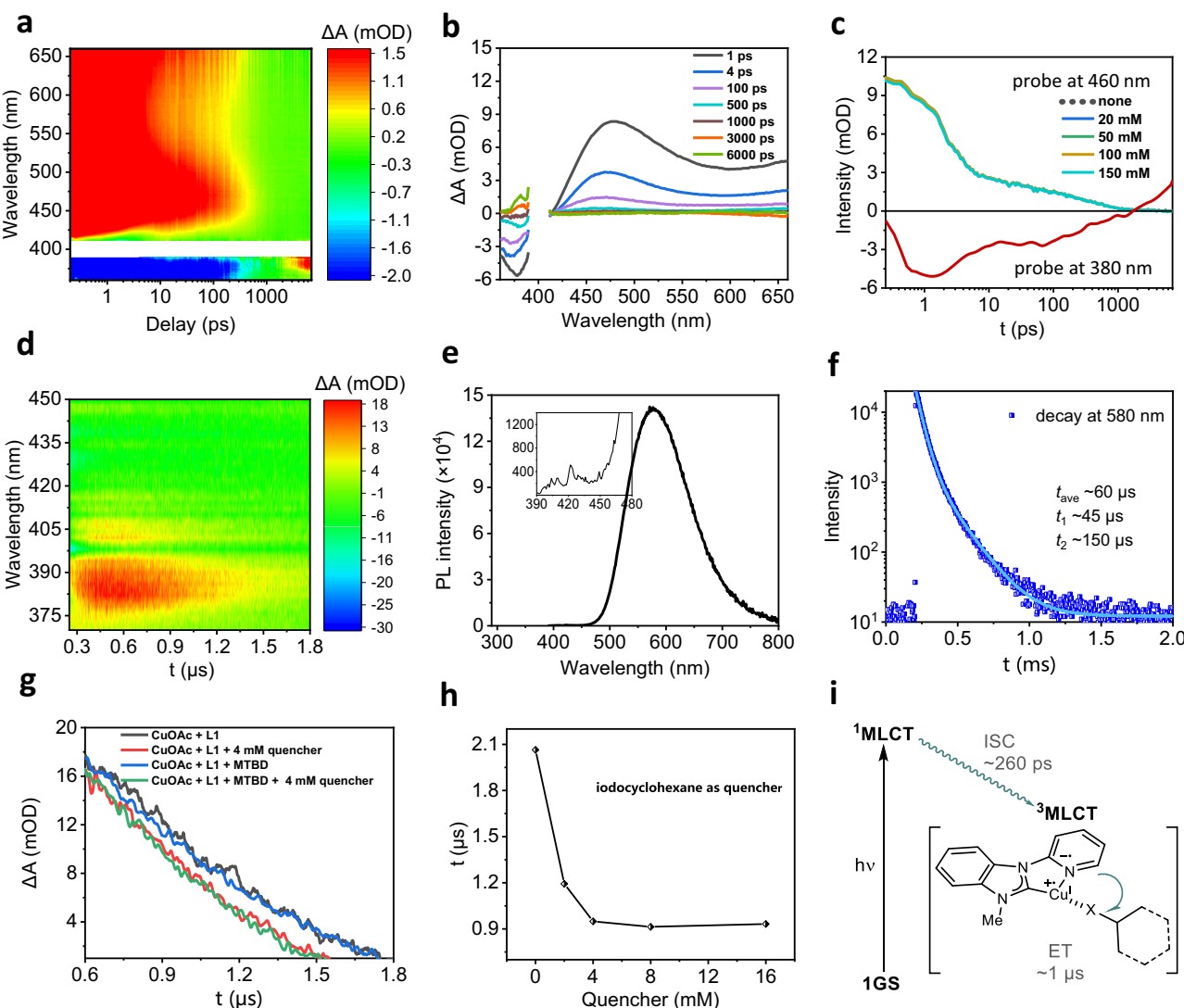

**Fig. 6 | Time-resolved spectroscopy studies. a** Two-dimensional pseudo-color femtosecond TA spectra of the in-situ generated **Cu-1** complex ($\lambda_{pump}$ = 400 nm). **b** Time-slicing TA spectra at selected pump-probe delays. **c** Single-wavelength TA ($\Delta$A) kinetics probed at 460 nm (upper panel) and at 380 nm (lower). The kinetics at 460 nm in the presence of varying concentrations of quencher (iodocydohex-ane) are also shown for comparison. **d** Two-dimensional pseudo-color nanosecond TA spectra of the in-situ generated **Cu-1** complex ($\lambda_{pump}$ = 355 nm). **e** Steady-state emission spectra of the in-situ generated **Cu-1** complex (excitation at 375 nm,

80 K). **f** Long-lived luminescence decay for the in-situ generated **Cu-1** complex measured at 80 K (excitation at 375 nm). **g** TA kinetics at 380 nm of the in-situ generated **Cu-1** complex (gray line). TA kinetics in the presence of quencher (red), MTBD (blue) and quencher plus MTBD (green) are also shown for comparison. **h** TA lifetime plotted as a function of the quencher concentration. **i** Proposed scheme for the intersystem crossing of the **Cu-1** complex and its inner-sphere SET to attached alkyl halide.

calculations showed that the singly occupied molecular orbital resides on the aniline ligand rather than the copper center for $Cu^{II}$ complex (Fig. 5d). The results obtained are consistent with the direct formation of the C–N bond through the coupling of the anilidyl radical with the alkyl radical.

## Time-resolved spectroscopy studies

While we have revealed that the in-situ generated **Cu-1** complex is the active photocatalyst, the reaction mechanisms for generation of alkyl radical in details remain to be clarified. We find that the room-temperature emission of the **Cu-1** complex is very weak (with quantum yield <3%). This issue prohibits reliable measurements of emission quenching, a method commonly adopted to elucidate the mechanisms of photoredox catalysis. In order to understand the origin of weak emission and to uncover the excited-state dynamics, we applied transient absorption spectroscopy across timescales from sub-picoseconds to sub-milliseconds; see supplementary section 4 for

experimental details. Fig. 6a shows the TA spectra as a function of both wavelength and time in the range of sub-ps to ns following 400 nm excitation. The negative feature at <400 nm corresponds to the ground-state bleach (GSB) of the **Cu-1** complex, whereas the broad-band absorption from ~420 nm to >650 nm can be assigned to the excited-state absorption (ESA) of the photogenerated $^1$MLCT states. Importantly, the ESA shows rapid decay within 100 s of ps, which is accompanied by the formation of a new absorption feature centered at ~380–390 nm; see also time-slicing TA spectra at selected delay times in Fig. 6b. These processes can be clearly visualized on the associated TA kinetics plotted in Fig. 6c. The first decay process has a time constant of 2.5 ps, which likely arises from internal conversion, and the second decay with a time constant of 260 ps remains to be assigned.

We therefore further measured the TA spectra on the ns to sub-ms timescales (Fig. 6d) and found that the absorption feature at ~380–390 nm has a lifetime of a few μs. On the basis of this long lifetime, we suspect that this feature arises from the triplet excited-

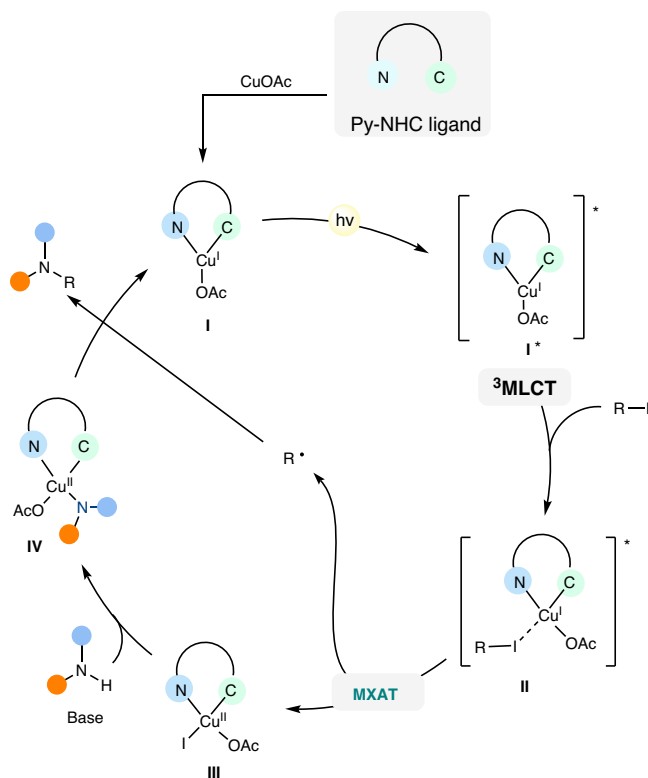

**Fig. 7 | Plausible Mechanism.** The generation of various species or intermediates in the proposed catalytic cycle.

of **Cu-1** could access the reduction potentials of unactivated alkyl halides. Based on cyclic voltammogram measurements and the phosphorescence spectrum, the oxidation potential of the ³MLCT state of **Cu-1** was estimated to be −2.19 V vs SCE[43], which was insufficient to reduce unactivated alkyl iodides ($E_{red} = -2.35$ V vs SCE) and bromides ($E_{red} = -2.75$ V vs SCE)[44]. If the alkyl halides are coordinated to ³MLCT state **Cu-1**, however, they could become activated to facilitate halide atom transfer to generate alkyl radicals. As the association between the in-situ **Cu-1** complex and the alkyl halide on the ground state were not observed by ¹H NMR and UV-vis spectra, we suspect that it occurs only for the photoexcited **Cu-1** complex, i.e., these two form exciplexes.

Based on the mechanistic studies, we propose a plausible mechanism as depicted in Fig. 7. The [Cuᴵ(Py-NHC)OAc] complex **I**, which is generated in situ, can be excited by 410 nm LEDs to form the ³MLCT excited state complex **I***. The **I*** associates with alkyl iodide to form an exciplex **II**, leading to halide atom transfer via inner-sphere SET to give an alkyl radical and Cuᴵᴵ complex **III**. The Cuᴵᴵ complex **III** undergoes ligand exchange to produce Cuᴵᴵ-amido complex **IV**. Consequently, complex **IV** react with the alkyl radical via radical coupling to generate the final product while simultaneously regenerating the copper complex **I**.

## Discussion

In summary, we have developed a copper photocatalytic system capable of facilitating C-N couplings between unactivated alkyl iodides or bromides and various N-nucleophiles, with excellent substrate scope and functional group compatibility. Notably, the developed process can be readily deployed for late-stage modification and scaled-up reactions (up to 6 mmol), thus demonstrating its practical applicability. We have further elucidated the underlying mechanisms, revealing that the formation of mono-pyridyl-carbene-ligated copper(I) complex accounts for the remarkable efficiency of halide atom transfer of alkyl halides via inner-sphere SET during C-N coupling reactions. As these findings have significant implications for both academic and industrial fields, they are expected to generate considerable interest and impact.

## Methods

### General procedure of C(sp3)-N couplings

To an oven-dried 10 mL reaction vial were added CuOAc (0.6 mg, 0.005 mmol, 5 mol%), **L1** (1.8 mg, 0.005 mmol, 5 mol%), and 1 mL MeCN in a nitrogen-filled glove box. The resulting mixture was stirred for 5 min, followed by adding 7-methyl-1,5,7-triazabicyclo[4.4.0]dec-5-ene (MTBD) (26 μL, 0.18 mmol, 1.8 equiv), N-nucleophiles (0.1 mmol, 1.0 equiv) and alkyl iodides/bromides (0.15 mmol, 1.5 equiv) in sequence, and sealed with a screwed cap. The sealed vial was placed on a photo-reactor under irradiation of LEDs (410 nm, 6 W). The mixture was stirred at 20−25 °C for 15 h, quenched with $H_2O$, and extracted with ethyl acetate. The combined organic layers were dried over anhydrous $Na_2SO_4$, and concentrated in vacuo. The crude product was purified by silica gel column chromatography to afford the coupling product.

## Data availability

All data that support the findings of this study are available within the paper and its supplementary information files, and also available from the corresponding author upon request. The X-ray crystallographic coordinates for structures reported in this study have been deposited at the Cambridge Crystallographic Data Centre (CCDC), under deposition numbers CCDC 224303. These data can be obtained free of charge from The Cambridge Crystallographic Data Centre via www.ccdc.cam.ac.uk/data_request/cif. Source data are provided with this paper.

## References

1. Aniszewski, T., *Alkaloids: Chemistry, Biology, Ecology, and Applications* (Elsevier, Amsterdam, 2015).

states (³MLCT). Indeed, at 80 K, we can clearly detect a strong emission band centered at 580 nm (Fig. 6e), which is ~100-fold stronger than the singlet emission at ~425 nm (Fig. 6e inset). The phosphorescence has a lifetime of ~60 μs at 80 K (Fig. 6f). At room temperature, it is shortened to a few μs due to activation of nonradiative recombination pathways, which nevertheless is sufficient for typical SET processes. Note that the GSB in Fig. 6a should remain for triplet states ³MLCT, but overlap of the GSB with the ESA of ³MLCT hides the former.

With the excited-state processes in the **Cu-1** complex uncovered, we further studied the excited-state quenching dynamics so as to derive the reaction mechanisms. To this end, we measured the TA dynamics in the presence of the reaction substrates (iodocyclohexane). As shown in Fig. 6c, the internal conversion and intersystem cross dynamics within 1 ns remain unchanged regardless of the concentration of iodocyclohexane added, which is expected as the SET processes can barely compete with these ultrafast processes. Therefore, charge transfer occurs for the triplet states. In Fig. 6g, we show the μs TA dynamics for the **Cu-1** complex in the presence of iodocyclohexane and/or MTBD. Clearly, addition of iodocyclohexane immediately shortens the TA lifetime, indicating that oxidative quenching of ³MLCT by iodocyclohexane is a viable pathway. In contrast, addition of MTBD does not obviously alter the TA dynamics, thus precluding the reductive quenching pathway. Consequently, simultaneous addition of both iodocyclohexane and MTBD has a similar effect as adding iodocyclohexane alone.

Importantly, when we plot the TA lifetime of the ³MLCT state of **Cu-1** as a function of the concentration of iodocyclohexane, we find a non-Stern-Volmer quenching behavior (Fig. 6h). The lifetime is shortened from 2.1 μs to 0.9 μs as the concentration of iodocyclohexane is raised from 0 to 4 mM, after which the lifetime remains constant. At 4 mM, the molar ratio between **Cu-1** and iodocyclohexane is approximately 1:20. This is an indication that interaction between **Cu-1** and iodocyclohexane is not diffusion-controlled but directly associated, which is actually the key to resolving a puzzle on how the ³MLCT state

2. Ricci, A. *Amino Group Chemistry: From Synthesis to the Life Sciences* (Wiley-VCH, 2008).

3. Vitaku, E., Smith, D. T. & Njardarson, J. T. Analysis of the structural diversity, substitution patterns, and frequency of nitrogen heterocycles among U.S. FDA approved pharmaceuticals. *J. Med. Chem.* **57**, 10257–10274 (2014).

4. Ruiz-Castillo, P. & Buchwald, S. L. Applications of palladium-catalyzed C-N cross-coupling reactions. *Chem. Rev.* **116**, 12564–12649 (2016).

5. Sambiagio, C., Marsden, S. P., Blacker, A. J. & McGowan, P. C. Copper catalysed Ullmann type chemistry: from mechanistic aspects to modern development. *Chem. Soc. Rev.* **43**, 3525–3550 (2014).

6. Dong, X.-Y., Li, Z. L., Gu, Q. S. & Liu, X. Y. Ligand development for copper-catalyzed enantioconvergent radical cross-coupling of racemic alkyl halides. *J. Am. Chem. Soc.* **144**, 17319–17329 (2022).

7. Zhang, Y. F. et al. Copper-catalyzed enantioconvergent radical C(sp3)-N cross-coupling: access to alpha,alpha-disubstituted amino acids. *Angew. Chem. Int. Ed.* **62**, e202302983 (2023).

8. Chen, J. J. et al. Copper-catalyzed enantioconvergent radical C(sp3)-N cross-coupling of activated racemic alkyl halides with (hetero)aromatic amines under ambient conditions. *J. Am. Chem. Soc.* **145**, 14686–14696 (2023).

9. Chen, J. J. et al. Enantioconvergent Cu-catalysed N-alkylation of aliphatic amines. *Nature* **618**, 294–300 (2023).

10. Fu, G. C. Transition-metal catalysis of nucleophilic substitution reactions: a radical alternative to S(N)1 and S(N)2 processes. *ACS Cent. Sci.* **3**, 692–700 (2017).

11. Hossain, A., Bhattacharyya, A. & Reiser, O. Copper's rapid ascent in visible-light photoredox catalysis. *Science* **364**, eaav9713 (2019).

12. Bissember, A. C., Lundgren, R. J., Creutz, S. E., Peters, J. C. & Fu, G. C. Transition-metal-catalyzed alkylations of amines with alkyl halides: photoinduced, copper-catalyzed couplings of carbazoles. *Angew. Chem. Int. Ed.* **52**, 5129–5133 (2013).

13. Do, H. Q., Bachman, S., Bissember, A. C., Peters, J. C. & Fu, G. C. Photoinduced, copper-catalyzed alkylation of amides with unactivated secondary alkyl halides at room temperature. *J. Am. Chem. Soc.* **136**, 2162–2167 (2014).

14. Matier, C. D., Schwaben, J., Peters, J. C. & Fu, G. C. Copper-catalyzed alkylation of aliphatic amines induced by visible light. *J. Am. Chem. Soc.* **139**, 17707–17710 (2017).

15. Ahn, J. M., Peters, J. C. & Fu, G. C. Design of a photoredox catalyst that enables the direct synthesis of carbamate-protected primary amines via photoinduced, copper-catalyzed N-Alkylation reactions of unactivated secondary halides. *J. Am. Chem. Soc.* **139**, 18101–18106 (2017).

16. Chen, C., Peters, J. C. & Fu, G. C. Photoinduced copper-catalysed asymmetric amidation via ligand cooperativity. *Nature* **596**, 250–256 (2021).

17. Liao, L., Song, L., Yan, S., Ye, J. & Yu, D. Highly reductive photocatalytic systems in organic synthesis. *Trends Chem.* **4**, 512–527 (2022).

18. Dow, N. W., Cabre, A. & MacMillan, D. A general N-alkylation platform via copper metallaphotoredox and silyl radical activation of alkyl halides. *Chem* **7**, 1827–1842 (2021).

19. Górski, B., Barthelemy, A., Douglas, J. J., Juliá, F. & Leonori, D. Copper-catalysed amination of alkyl iodides enabled by halogen-atom transfer. *Nat. Catal.* **4**, 623–630 (2021).

20. Mao, R., Frey, A., Balon, J. & Hu, X. Decarboxylative C(sp3)–N cross-coupling via synergetic photoredox and copper catalysis. *Nat. Catal.* **1**, 120–126 (2018).

21. Liang, Y., Zhang, X. & MacMillan, D. W. C. Decarboxylative sp3 C–N coupling via dual copper and photoredox catalysis. *Nature* **559**, 83–88 (2018).

22. Li, J., Huang, C. & Li, C. Two-in-one metallaphotoredox cross-couplings enabled by a photoactive ligand. *Chem* **8**, 2419–2431 (2022).

23. Hernandez-Perez, A. C. & Collins, S. K. Heteroleptic Cu-based sensitizers in photoredox catalysis. *Acc. Chem. Res.* **49**, 1557–1565 (2016).

24. Beaudelot, J. et al. Photoactive copper complexes: properties and applications. *Chem. Rev.* **122**, 16365–16609 (2022).

25. Lin, C. Y., Coote, M. L., Gennaro, A. & Matyjaszewski, K. Ab initio evaluation of the thermodynamic and electrochemical properties of alkyl halides and radicals and their mechanistic implications for atom transfer radical polymerization. *J. Am. Chem. Soc.* **130**, 12762–12774 (2008).

26. Fang, C. et al. Mechanistically guided predictive models for ligand and initiator effects in copper-catalyzed atom transfer radical polymerization (Cu-ATRP). *J. Am. Chem. Soc.* **141**, 7486–7497 (2019).

27. Ji, C., Han, J., Li, T., Zhao, C., Zhu, C. & Xie, J. Photoinduced gold-catalyzed divergent dechloroalkylation of gem-dichloroalkanes. *Nat. Catal.* **5**, 1098–1109 (2022).

28. Chen, X. et al. Rational design of strongly blue-emitting cuprous complexes with thermally activated delayed fluorescence and application in solution-processed OLEDs. *Chem. Mater.* **25**, 3910–3920 (2013).

29. Cheung, K. P. S., Sarkar, S. & Gevorgyan, V. Visible light-induced transition metal catalysis. *Chem. Rev.* **122**, 1543–1625 (2022).

30. Cramer, J., Sager, C. P. & Ernst, B. Hydroxyl groups in synthetic and natural-product-derived therapeutics: a perspective on a common functional group. *J. Med. Chem.* **62**, 8915–8930 (2019).

31. Du, X. et al. Copper-catalyzed enantioconvergent radical N-Alkylation of Diverse (Hetero)aromatic Amines. *J. Am. Chem. Soc.* **146**, 9444–9454 (2024).

32. Zheng, J. et al. Copper-catalyzed enantioconvergent radical C(sp3)–N cross-coupling to access chiral α-Amino-β-lactams. *Precis. Chem.* **1**, 576–582 (2023).

33. Zhang, Y. et al. Enantioconvergent Cu-catalyzed radical C–N coupling of racemic secondary alkyl halides to access α-chiral primary amines. *J. Am. Chem. Soc.* **143**, 15413–15419 (2021).

34. Zuccarello, G., Batiste, S. M., Cho, H. & Fu, G. C. Enantioselective synthesis of α-aminoboronic acid derivatives via copper-catalyzed N-Alkylation. *J. Am. Chem. Soc.* **145**, 3330–3334 (2023).

35. Kainz, Q. M. et al. Asymmetric copper-catalyzed C-N cross-couplings induced by visible light. *Science* **351**, 681–684 (2016).

36. Stanton, C. J.III et al. Re(I) NHC complexes for electrocatalytic conversion of $CO_2$. *Inorg. Chem.* **55**, 3136–3144 (2016).

37. Garrison, J. C. & Youngs, W. J. Ag(I) N-Heterocyclic Carbene Complexes: synthesis, structure, and application. *Chem. Rev.* **105**, 3978–4008 (2005).

38. Lake, B. R. M. & Willans, C. E. Structural diversity of Copper(I)–N-Heterocyclic carbene complexes; ligand tuning facilitates isolation of the first structurally characterised Copper(I)–NHC containing a Copper(I)–Alkene interaction. *Chem. Eur. J.* **19**, 16780–16790 (2013).

39. Cho, H., Suematsu, H., Oyala, P. H., Peters, J. C. & Fu, G. C. Photoinduced, copper-catalyzed enantioconvergent alkylations of anilines by racemic tertiary electrophiles: synthesis and mechanism. *J. Am. Chem. Soc.* **144**, 4550–4558 (2022).

40. Jang, E. S., McMullin, C. L., Käß, M., Meyer, K., Cundari, T. R. & Warren, T. H. Copper(II) Anilides in sp3 C–H Amination. *J. Am. Chem. Soc.* **136**, 10930–10940 (2014).

41. Jayasooriya, I. U. et al. Copper(ii) ketimides in sp3 C–H amination. *Chem. Sci.* **12**, 15733–15738 (2021).

42. Lee, H. et al. Investigation of the C–N bond-forming step in a photoinduced, Copper-catalyzed enantioconvergent N–Alkylation: characterization and application of a stabilized organic radical as a mechanistic probe. *J. Am. Chem. Soc.* **144**, 4114–4123 (2022).

43. Buzzetti, L., Crisenza, G. E. M. & Melchiorre, P. Mechanistic studies in photocatalysis. *Angew. Chem. Int. Ed.* **58**, 3730–3747 (2019).
44. Constantin, T., Zanini, M., Regni, A., Sheikh, N. S., Juliá, F. & Leonori, D. Aminoalkyl radicals as halogen-atom transfer agents for activation of alkyl and aryl halides. *Science* **367**, 1021–1026 (2020).

## Acknowledgements

The authors acknowledge financial support from the National Natural Science Foundation of China (No. 22371027 for L.L.). K.W. acknowledges financial support from the Chinese Academy of Sciences (YSBR-007) and the New Cornerstone Science Foundation through the XPLORER PRIZE.

## Author contributions

L.L. conceived the research project. L.L. and H.L. designed the catalytic reactions. H.L. contributed to optimizing the coupling reactions and mechanistic studies. Y.Y. performed experiments of TA mechanistic studies. Y.F. performed the calculation. H.L., F.Y., L.G., and Y.M. performed experiments of catalytic reactions or/and ligand synthesis. L.L., K.W., and Y.L. co-directed the research project. L.L., K.W., Y.L, H L., and Y.Y. prepared the manuscript. All authors contributed to discussions and commented on the manuscript.

## Competing interests

The authors declare no competing interests.
