## [Peer Review File · Nature Communications]

In situ copper photocatalysts triggering halide atom transfer of unactivated alkyl halides for general C(sp³)-N couplingsReviewers' Comments:

Reviewer #1:

Remarks to the Author:

[Note from the Editor: Reviewer #1 was asked to assess the response given to reviewer #2 who was not able to look over the revision again.]

The work effectively demonstrates the efficacy of the developed copper catalyst in enabling general C(sp³)-N couplings across a broad range of substrates. The standardized reaction conditions accommodate various nitrogen nucleophiles along with unactivated alkyl iodides/bromides. The ability to modify complex molecules at a late stage and perform scale-up reactions showcases the practicality of the method. The detailed mechanistic investigations, particularly the application of transient absorption spectroscopy to elucidate the formation of alkyl radicals via copper exciplex-induced inner-sphere electron transfer, are noteworthy and hold promise for advancing metal catalysis. Given the discovery of the novel photocopper catalytic system and process, I think this work is potentially suitable for publication in Nature Communications. I have already checked the responses to reviewers and the revised manuscript, and I think the authors have addressed all of concerns of the reviewers. However, a few minor points about this manuscript should be further addressed.

First, the authors mentioned general C(sp³)-N couplings in manuscript, but the secondary amines would not react as the nucleophiles. I think the authors should give some explanation for that.

Next, I am curious about generation of alkyl radical via copper-exciplex associated with alkyl halides lead to selective coupling between alkyl halide with amines. The redox potential of the excited-state copper complex is not sufficient to reduce the unactivated alkyl halides via outer-sphere SET process. How about the reactivity of redox-active esters as the coupling partners which match the redox potential? If they work well, the method would be much more useful. I think the authors need more experimental data to clarify that.

Finally, based on the proposed mechanism, some chiral Py-NHC ligands would be possible to lead to enantioselective version. I recommend author to try some chiral ligands, which would be very nice to know the possibility, even the primary enantioselectivity is not very good.

Response to reviewers

Reviewer #1 (Remarks to the Author):

The work effectively demonstrates the efficacy of the developed copper catalyst in enabling general C(sp³)-N couplings across a broad range of substrates. The standardized reaction conditions accommodate various nitrogen nucleophiles along with unactivated alkyl iodides/bromides. The ability to modify complex molecules at a late stage and perform scale-up reactions showcases the practicality of the method. The detailed mechanistic investigations, particularly the application of transient absorption spectroscopy to elucidate the formation of alkyl radicals via copper exciplex-induced inner-sphere electron transfer, are noteworthy and hold promise for advancing metal catalysis. Given the discovery of the novel photocopper catalytic system and process, I think this work is potentially suitable for publication in Nature Communications. I have already checked the responses to reviewers and the revised manuscript, and I think the authors have addressed all of concerns of the reviewers. However, a few minor points about this manuscript should be further addressed.

[Our response] Many thanks for your positive comments on this work.

[Comments 1#] First, the authors mentioned general C(sp³)-N couplings in manuscript, but the secondary amines would not react as the nucleophiles. I think the authors should give some explanation for that.

[Our response] Thank you for your comments. One plausible explanation is that the higher basicity of secondary amines poses a greater challenge for the deprotonation process, which is essential for coordinating with metals through ligand substitution. This also explains why coupling with primary amines to produce secondary amines is possible without generating over N-alkylation side products.

[added comments in the updated manuscript]

to form the desired products (24-25), and over-alkylation products were not observed. Secondary amines were ineffective, possibly because of the slow deprotonation process hindering their coordination with copper via ligand substitution under the current conditions. (Supplementary Figure 24). ↵

[Comments 2#] Next, I am curious about generation of alkyl radical via copper-exciplex associated with alkyl halides lead to selective coupling between alkyl halide with amines. The redox potential of the excited-state copper complex is not sufficient to reduce the unactivated alkyl halides via outer-sphere SET process. How about the reactivity of redox-active esters as the coupling partners which match the redox potential? If they work well, the method would be much more useful. I think the authors need more experimental data to clarify that.

[Our response] Thank you for your comments and suggestions. We attempted to use electrophiles with higher redox potential, such as redox esters; however, we did not achieve the desired C-N coupling product. This outcome suggests that the halogenophilicity of the excited-state copper is a crucial factor for the success of this reaction.

[added comments in the revised manuscript]

coupling of alkyl chlorides with N-nucleophiles proved unsuccessful under the present conditions. However, other electrophiles with higher redox potential, such as redox esters, are incompatible (Supplementary Figure 27).⁴

[Comments 3#] Finally, based on the proposed mechanism, some chiral Py-NHC ligands would be possible to lead to enantioselective version. I recommend author to try some chiral ligands, which would be very nice to know the possibility, even the primary enantioselectivity is not very good.

[Our response] Thank you for your comments and suggestions. We have successfully synthesized a simple chiral Py-NHC ligand and conducted a C-N coupling reaction (Supplementary Figure 28). Unfortunately, we did not achieve positive results. Asymmetric C-N coupling is a crucial research area, as referenced in (6-9, 16, 31-36), demanding the development of more efficient ligands to differentiate between the two similar methylene groups in the radical addition to metal centre step. Although we did not achieve the desired outcome in this study, the development of the chiral ligand is currently ongoing in our laboratory. We will share the relevant results once we have gathered a sufficient number of ligands, although introducing chiral structures to the planar Py-NHC ligands is not a trivial task.

[added comments in the revised manuscript]

3b). Attempts to achieve enantioselective C-N couplings^{6-9, 16, 31-36} were unsuccessful with chiral Py-NHC chiral ligands, possibly because of the lack of efficient chiral ligands to differentiate two similar methylene groups. (Supplementary Figure 28).⁴

Thank you for your valuable feedback and inspiration for future research directions. Your comments are greatly appreciated!

Reviewers' Comments:

Reviewer #1:

Remarks to the Author:

In the revised version of the manuscript, the authors have made a great effort to address the concerns of myself through additional experiments. Now, this manuscript is suitable for publication in NC.

Response to reviewer and editor

Reviewer #1 (Remarks to the Author):

In the revised version of the manuscript, the authors have made a great effort to address the concerns of myself through additional experiments. Now, this manuscript is suitable for publication in NC.

[Our response] Many thanks for your supportive comments.

Editor #1 (Remarks to the Author):

You will need to upload:

1. Editorial Policy Checklist
2. Completed Third Party Rights Table (if relevant)
3. A completed copy of this checklist 1 file
4. The main manuscript file in either Microsoft Word or LaTeX format
5. Separate Figure files

[Our response] Thank you for handling our manuscript. we have updated the formatting of the manuscript and have successfully uploaded all the necessary files.